# The Role of Convection and Size Effects in Microhotplate Heat Exchange: Semiconductor and Thermomagnetic Gas Sensors [note 1]

**DOI:** 10.3390/s25092830

**Published:** 2025-04-30

**Authors:** Alexey Vasiliev, Alexey Shaposhnik, Oleg Kul, Artem Mokrushin

**Affiliations:** 1Laboratory of Sensor Systems, Dubna State University, Universitetskaya Strt., 19, 141980 Dubna, Russia; 2LLC “C-Component”, Tushinskaya Strt., 17, 125362 Moscow, Russia; okul@c-component.ru; 3Department of Chemistry, Voronezh State Agrarian University, Michurin Strt., 1, 394087 Voronezh, Russia; a.v.shaposhnik@gmail.com; 4Kurnakov Institute of General and Inorganic Chemistry, Russian Academy of Sciences, Leninsky Av. 31, 119991 Moscow, Russia; artyom.nano@gmail.com

**Keywords:** microheater, convective heat losses, thermal conductivity, Grashof number

## Abstract

The analysis of the influence of microhotplate size on the convective heat exchange of gas sensors is presented. Usually, the role of convection in the heat exchange of gas sensors is not considered in thermal simulation models because of the complexity of the convection process. As a result, the contribution of this process to the overall heat loss of sensors remains without detailed analysis. We analyzed convection issues in two groups of gas sensors: semiconductor and thermocatalytic (calorimetric) sensors and, on the other hand, in the oxygen sensors of the thermomagnetic type. It is demonstrated that there is a critical size leading to the formation of convective heat exchange flow. Below this critical value, only thermal conductivity of ambient air, IR (infrared) radiation from the heated microhotplate surface, and thermal conductivity of the microhotplate-supporting elements should be considered as channels for heat dissipation by the microhotplate, and the contribution of free convection can be neglected. The expression for the critical size contains only fundamental constants of air, d_cr_~4·ν·Dg3, where ν is the kinematic viscosity of air, D is the diffusion coefficient, and g is the acceleration of free fall, d_cr_~0.5 cm. Therefore, if the size of the microhotplate d <<d_cr_, the influence of convection heat exchange can be neglected. Similar results were obtained in the analysis of the behavior of thermal magnetic sensors of oxygen, which use paramagnetic properties of molecular oxygen for the determination of O_2_ concentration. In this case, the critical size of the sensor is also of significance; if the size of the magnetic sensor is much below this value, the oxygen concentration value measured with such a device is independent of the orientation of the sensor element. The results of the simulation were compared with the measurement of heat loss in micromachined gas sensors. The optimal dimensions of the sensor microhotplate are given as a result of these simulations and measurements.

## 1. Introduction

Sensors of different physical and chemical parameters of air (or of a gas, in general) have recently offered a variety of important online information about the systems that control the environment, technological processes, and atmosphere in residential and office buildings. Sensors of airflow, pressure, chemical composition and impurities in the air are based on different physical and chemical principles. However, many such sensors rely on the temperature difference between elements of the sensor as a main measurement parameter, ultimately providing information about the target value characterizing the ambient environment.

Among these sensors are the following: thermal anemometers, which measure gas flow velocity using the temperature difference between hotplates placed in a series within the gas flow [1]; thermocatalytic (calorimetric) sensors, which measure temperature difference between two hot elements coated and uncoated with a catalyst, active in the process of surface oxidation of target combustible gas [2]; and magnetic oxygen sensors, which measure gas flow rate of oxygen-containing gases using hot elements placed in a magnetic field. This flow arises due to the paramagnetic properties of oxygen, the only paramagnetic gas commonly present in the atmosphere (aside from relatively rare NO) [3].

Another group of sensors includes semiconductor gas sensors [4,5], which use a single hotplate to maintain a desirable high temperature required for fast catalytic oxidation of the target gas on the surface of the semiconducting catalytic nanomaterial, and Pirani pressure gauges [6,7]. In these sensors a microhotplate serves as a heating element with either stabilized (in semiconductor gas sensors) or measured (in Pirani gauges) temperature.

A current trend in the fabrication of all these sensors is the application of microhotplates made using one of the possible microelectronic technologies, such as silicon MEMS, thin ceramic alumina or LTCC (low temperature co-fired ceramic) membranes [8], screen-printed glass-ceramic membranes or cantilevers [9], and others. The main reason for using microfabrication technologies, aside from the optimization of sensor cost, is the minimization of the power consumption of the sensor’s hotplate.

In order to minimize the power consumption of these microhotplates, it is necessary to analyze the paths of heat loss. This analysis is relatively simple in the case of losses due to radiation or thermal conduction, but an analysis of convection is always problematic because of the complex characteristics of this process. In this paper, we formulate criteria enabling the contribution of convection to the heat losses of a sensor operating at high temperature to be neglected.

## 2. Heat Dissipation by Microhotplates

A microhotplate dissipates heat in four ways (Figure 1):(1)Conduction through supporting structures, such as wires suspending the hotplate, or a membrane supporting the microhotplate in the case of MEMS-based sensor elements;(2)Conduction through ambient gas, resulting from the thermal conductivity of the surrounding medium;(3)Natural thermal convection occurring in the air gap between the microhotplate and the sensor housing;(4)Thermal radiation emitted by the surface of the microhotplate.

Sensor radiation losses are negligible under typical sensor dimensions and operating temperatures. Indeed, the radiation loss from the sensor microhotplate obeys the Stefan–Boltzmann law, W = σ∙T^4^∙s, where W—power dissipated due to radiation; σ—Stefan–Boltzmann constant; σ = 5.67 × 10^−8^ W∙m^−2^∙K^−4^; T—absolute temperature of the sensor; and s—the area of the sensor. Under typical conditions, s is of the order of 1 mm^2^ and T = 700 K. Taking into account these values, radiation heat losses can be estimated as W~2 mW. This value is significantly lower than the power consumption of microhotplates commonly used in, for example, methane sensors operating at 450 °C, where the power consumption is approximately 200 mW [8].

In these estimations, the microhotplate is assumed to behave as a black-body emitter. In reality, its emissivity is lower, so the actual radiation losses are even smaller than this maximum theoretical estimate.

Understandings of the thermal conductivity of air and of the sensor elements are more or less clear. The heat losses related to these effects are proportional to the temperature difference between the microhotplate and the housing of the sensor. Heat loss due to the thermal conductivity of the supporting elements is independent of the microhotplate area, while loss due to the thermal conductivity of air is proportional to the area.

For example, as demonstrated in [8], in membrane-type microhotplates consisting of a platinum microheater on 1.2 μm thick SiO_2_/Si_3_N_4_ dielectric membrane, heat losses exhibit a monotonic dependence on membrane size. With a fixed size of the microheater itself equal to 250 × 250 μm, the thermal losses decrease as the membrane size increases, eventually reaching a minimum constant value. This saturation occurs when the ratio of membrane size to microheater size reaches approximately 8. Further increase in membrane size doesn’t reduce heat losses but significantly increases the membrane’s fragility.

Based on both simulation results and experimental data, the optimal size of the membrane supporting a 250 × 250 μm microheater was determined to be approximately 1.5 × 1.5 mm.

In contrast to heat conduction and radiation losses, the role of convection in the heat dissipation of the microheater has, to the best of our knowledge, never been thoroughly investigated. This remains a common and unresolved issue in the design of low power-consuming semiconductor and thermocatalytic (calorimetric) gas sensors based on microhotplates [10,11], as well as other sensors working at elevated temperatures, such as thermoconductometric sensors of gas concentration [12], airflow sensors, magnetic sensors of oxygen concentration, etc.

In principle, convection could be important, because it may not only increase total heat loss and, consequently, power consumption, but can also introduce orientation-dependent effects in the sensor’s response. In most thermal simulations, however, only heat conduction through the air and through the supporting structures of the microhotplate is considered, while radiation is typically neglected, as previously discussed.

Yet, clear estimates of the minimum microhotplate size required for the onset of natural convection are rarely found in the literature.

From an intuitive standpoint, if the microhotplate is extremely small, essentially acting as a hot point, then natural convection is unlikely to be initiated. In such a case, the buoyant (Archimedean) force acting on the heated air above the microhotplate is insufficient to generate convective flow. On the other hand, it is also clear that, if the temperature difference between the microhotplate and housing is too small, the convection also cannot start. Therefore, in our consideration, we should take into account these two boundary conditions.

In this work, we aim to estimate the critical size below which the contribution of convection to total heat loss can be neglected. This threshold is essential for optimizing the design of gas sensors that utilize microhotplates and operate under high-temperature conditions.

Another issue, which should be taken into account, is the optimization of sensor microhotplate size from both technological and physical points of view. Currently, the majority of metal oxide semiconductor and thermocatalytic gas sensors are fabricated using sensing layers prepared by sol–gel, plasma, or other technologies, giving this material in powder form. This approach offers higher stability and higher responses of such materials compared to sputtered ones. Therefore, the sensing material should be deposited onto the microhotplate in the form of ink or paste, and this restricts minimum size of the microhotplate by a value of about 200 μm.

On the other hand, the size of the total membrane supporting the sensing layer should be chosen by taking into account the minimization of power consumption at the appropriate working temperature, for example, at 450 °C for methane detection. This balance between minimizing power consumption and ensuring effective sensor performance is crucial.

The most comprehensive classical consideration of all aspects of the heat and mass transfer processes and their influence onto the kinetics of chemical reactions was presented many years ago in the book [13]. While this work provides a broad overview of thermal and mass transfer considerations, the discussion of convection is relatively brief. Unfortunately, this analysis does not provide significant insight into the specific role of convection in heat exchange within microhotplates. In fact, the author only mentions a dimensionless parameter, which can be related to convection, the so called Grashof number:$$Gr=\frac{g\cdot L^{3}\cdot \beta \cdot \left(T-T_{r}\right)}{v^{2}}$$
where *g* is the gravitational acceleration, *L* is the typical geometrical size of the surface, *T* is the temperature of the hot surface, *T_r_* is the ambient temperature, ν is the kinematic viscosity of air, and β is the coefficient taking into account the heat exchange process on the surface.

This number was constructed to form a dimensionless value, taking into account all parameters, which could control the process of convection.

Unfortunately, there is no recommendation on how to use this number for the analysis of heat transfer from microhotplates.

In the sensor community, there are some legends, which are not confirmed by reference to any published documents. For example, we found an opinion that the authors of the book [14] analyzed this problem and found the criteria (the critical size of the hot element), which permits the neglecting of free convection. In fact, the authors of this book analyzed the operation of thermocatalytic sensors based on platinum spirals and supposed that the small dimension of the spiral, equal to ~100 μm, permitted them not to consider the influence of the free convection of air in housing as a mechanism of cooling the microsensor.

## 3. Convection in Thermomagnetic Sensors of Oxygen

A similar problem arises in the design and analysis of the operation of magnetic sensors of oxygen.

The operation principle of magnetic sensors of oxygen is based on paramagnetic properties of molecular oxygen [15]. There are two different approaches to the application of magnetic properties of oxygen for the measurement of its concentration.

The first one is the use of bulk properties of oxygen as paramagnetic gas and the use of its attraction by the magnetic field [15]. The second and, perhaps more interesting, is the application of the so-called Senftleben effect [16,17,18]. The Senftleben effect consists in a change in the diffusion cross-section of the triplet-state oxygen molecules in the magnetic field. The reason for this increase in the cross-section is the precession of oxygen molecules in the magnetic field. Therefore, in magnetic field, the coefficients of temperature conductivity and thermal conductivity decrease, and this change can be detected by measuring the variation in microhotplate temperature in the pulsing magnetic field.

This effect depends on the ratio of magnetic-field strength to gas pressure. Therefore, this effect is valuable at a low pressure of 0.01–2 Torr. The threshold of O_2_ detection is very low and is about 10^−5^ Torr. Due to its high sensitivity, this effect is particularly valuable for leak detection in vacuum systems.

However, in the present work, we focus primarily on more conventional devices that exploit the bulk magnetic properties of gases containing paramagnetic components.

The contemporary state of the art in the field of magnetic sensors of oxygen concentration is presented, for example, in a review article [15].

The paramagnetic properties of oxygen are related to the electronic configuration of the oxygen molecule. It is known that according to Hund’s law, electrons in oxygen molecules occupying π-orbital first occupy the state with parallel spins. Therefore, the total magnetic moment of the oxygen molecule is unity, and, as a result, it has a magnetic moment in contrast to almost all common gases in the atmosphere, except nitrogen oxide. The magnetic moment of oxygen exceeds the moment of all other gases by a factor of 100, and this property enables selective detection of oxygen in the presence of other gases.

On the other hand, this property imposes limitations on the minimum detectable concentration of oxygen. As a result, magnetic oxygen sensors are predominantly employed for industrial monitoring and medical diagnostics, where oxygen concentrations are relatively high and comparable to those found in ambient air.

The magnetic susceptibility of oxygen follows Curie’s law, being inversely proportional to absolute temperature. Therefore, the interaction of oxygen with the magnetic field is much stronger at a low temperature; this property enables the fabrication of the most simple and the most widely usable instrument [15]. The scheme of this instrument is presented in Figure 2, and in our work we will analyze mostly the instruments of this type.

The operating principle of the sensor based on the so-called “magnetic wind” effect is illustrated in the schematic diagram and description provided in Figure 2. The sample gas passes through two symmetrical sections of the ring-shaped chamber, designed to balance the pressure and velocity at the inlet and outlet of a glass (diamagnetic) tube connecting these sections. A wire coil is wound around this tube and forms two arms of a Wheatstone bridge used both for gas heating and for the measurement of its velocity; these coils work as a thermal anemometer.

The working principle of this instrument is based on a decrease in magnetic susceptibility of oxygen with an increase in temperature (Curie’s law). Consequently, the magnetic susceptibility of gas on the left end of the glass tube is higher than the susceptibility on the right side, and this creates the macroscopic gas motion from left to right. The velocity of this flow is directly related to the oxygen concentration in gas.

The key problem restricting the limit of detection of such a sensor is the competing influence of the free convection of gas. If the sensor tube is not perfectly horizontal, gas can move in the same (or opposite) direction due to unforced convection. The analysis presented in [19] demonstrates that the limit of detection of such sensors can be of the order of 100 ppm of oxygen in a diamagnetic background gas.

Therefore, it is crucial to determine the optimal geometry of thermal magnetic oxygen sensors that would eliminate or minimize the influence of free thermal convection, ensuring that measurement results are independent of sensor orientation. Achieving such immunity to orientation effects would significantly improve the sensor’s applicability in portable devices. Moreover, miniaturization of the sensor design would reduce the characteristic diffusion time and make the sensor applicable for the monitoring of oxygen in real-time breath analysis, even in sport medicine, where the quantification frequency should be up to 100 Hz.

The primary objective of this study is to investigate the influence of microhotplate and magnetic sensor geometry on the effects of free thermal convection, and to establish design criteria that allow its contribution to be reliably neglected.

## 4. Methods and Analysis

### 4.1. Convection Effects in Semiconductor and Thermocatalytic Sensors

As noted previously, there is a common consensus that, if a microhotplate with a size of a few tens of hundreds of microns is used as a heater of a semiconductor or thermocatalytic gas sensor, it is possible not to take into account the influence of free thermal convection on the heat exchange processes of gas sensors. This conclusion stems both from available experimental observations and from the inherent complexity of modeling convective processes. However, it remains difficult to find rigorous justification for the assumption that convection is negligible in systems involving micro-scale heating elements.

For the evaluation of the minimum size of microhotplate capable of inducing an ascending (buoyant or flowing up in gravitation field) air flow from the hotplate to, for instance, the colder wall of the sensor housing, we formulated a model based on the following assumptions.

(1)There is competition between two processes: the ascending air flow due to free thermal convection and the back-diffusion of air molecules. If the convection of gas plays an important role in heat transfer, the characteristic time of convective transfer should be shorter (or, preferably, much shorter) than that of diffusion. If the diffusion (it is known that the diffusion coefficient in ideal gas is equal to the temperature conductance coefficient) of gas is faster than the convection flow, the role of convection is negligible, and all heat exchange processes are only due to the process of heat conductivity.(2)The convective flow in a virtual convection tube is laminar, because the velocity of this flow is very small, and the Reynolds number Re = v∙L/ν, where v—average velocity of gas flow, v < 10 cm/s, L—characteristic size of the sensor, L < 1 cm, and ν—kinematic viscosity of air, ν~1 cm^2^/s. Therefore, Re < 10, and the flow is in the deep laminar zone. A virtual convection tube is a tube placed over the hot surface with gas flowing up due to free convection.(3)Air in this ascending flow (convection tube), heated by a microheater, is then cooled by surrounding room-temperature air due to the heat conductance of air along the sides of the virtual convection tube.(4)The result should meet the criteria of reasonability: under isothermal conditions, the convection is suppressed, and, for a point-like heat source, the Archimedes’ force is insufficient to establish a stable upward flow.

Let us consider a microhotplate with diameter d and, respectively, radius r = d/2 (Figure 3). We suppose that there is a convection tube over this microhotplate, the temperature of lower end of this tube is equal to the temperature of the microhotplate T, and the temperature of the upper end is equal to the room temperature T_r_. The height of the tube is equal to h. The convection velocity of gas in the tube is v. The condition, which should be met to neglect convection is t > τ, or, better, t >> τ (t—is convection time, τ—diffusion time) orh/v > h^2^/D, (1)
where D is diffusion coefficient.

The gas flow in the convection tube is laminar; therefore, there is no gas mixing, and the cooling time of the gas in the tube is equal to τ_1_ = r^2^/D. As a result, the height of the tube, necessary for the cooling of gas in it, is equal to h = v·τ_1_ = v·r^2^/D. This is the height of the diffusion tube in Figure 3. Formula (1) can be rewritten as(2)$$\frac{v\cdot r}{D}<1$$

The velocity of gas in the convection tube can be evaluated using the Poiseuille formula$$Q=\frac{\pi \cdot r^{4}}{8\cdot \eta \cdot h}\cdot \Delta P,\;\eta =\nu \cdot \rho ,\;Q=\pi \cdot r^{2}\cdot v,$$
where ν—kinematic viscosity of air, η—dynamic viscosity, ρ—air density, Q—gas flow, v—linear velocity of gas, and ΔP—pressure difference on the ends of the convection tube.

Therefore, gas velocity in the convection tube is equal to(3)$$v=\frac{r^{2}}{8\cdot \eta \cdot h}\cdot \Delta P.$$

Let us try to estimate the Archimedes’ force acting on the gas in the convection tube. The mass of the gas over the microheater at room temperature *T_r_* and at high temperature *T* are equal to, respectively,$$m_{r}=\frac{\mu \cdot P\cdot V}{R\cdot T_{r}}=\frac{\mu \cdot P\cdot \pi \cdot r^{2}\cdot h}{R\cdot T_{r}}$$
and$$m_{T}=\frac{\mu \cdot P\cdot V}{R\cdot T_{r}}=\frac{\mu \cdot P\cdot \pi \cdot r^{2}\cdot h}{R\cdot T}$$

The Archimedes’ force acting on the air in the convection tube is equal to the difference between the weights of hot and cold air in the convection tube:$$F=g\cdot (m_{r}-m_{T})=\frac{g\cdot \mu \cdot P\cdot \pi \cdot r^{2}\cdot h}{R}(\frac{1}{T_{r}}-\frac{1}{T}),$$
where μ—mass of mole of air, P—ambient pressure, and R—universal gas constant. Therefore, the Archimedes’ pressure is of$$\Delta P=\frac{F}{\pi \cdot r^{2}}=\frac{g\cdot \mu \cdot P\cdot h}{R}(\frac{1}{T_{r}}-\frac{1}{T})$$

We can rewrite this expression (3) as(4)$$v=\frac{r^{2}}{8\eta \cdot h}\cdot \Delta P=\frac{g\cdot \mu \cdot P\cdot h\cdot r^{2}}{8\eta \cdot h\cdot R}\cdot (\frac{1}{T_{r}}-\frac{1}{T})=\frac{g\cdot \mu \cdot P\cdot r^{2}}{8\eta \cdot R}\cdot (\frac{1}{T_{r}}-\frac{1}{T})$$

We should account for the ideal gas laws and relationship between kinematic and dynamic viscosity. As a result, the expression (4) can be written as(5)$$v=\frac{g\cdot r^{2}\cdot T_{av}}{8\nu }(\frac{1}{T_{r}}-\frac{1}{T}),$$
where T_av_ is a certain average temperature in the convection tube.

Substituting this expression into Formula (2), it is possible to obtain the following limit for the microhotplate.(6)$$\frac{g\cdot r^{3}}{8\nu \cdot D}\cdot (\frac{T_{av}}{T_{r}}-\frac{T_{av}}{T})<1\mathrm{or}\;r<\sqrt[{3}]{\frac{8\cdot \nu \cdot D}{g\cdot (\frac{T_{av}}{T_{r}}-\frac{T_{av}}{T})}}.$$

For usual methane sensors, for example [8], the value of temperatures used in expression (6) are T_r_ = 300 K, T = 750 K, and T_av_ = 525 K. Therefore, the term in brackets is equal to 1.05. However, if the temperature of the microhotplate is very low, close to room temperature, the critical size of the microhotplate is very big, and convection is not observed more or less at any size of hotplate according to the reasonable boundary condition of this problem.

However, for usual conditions and usual gas sensor microhotplates, the cubic route of the term in brackets is close to unity; therefore, generally, the convection part can be neglected, if the following condition is met:$$d<4\cdot \sqrt[{3}]{\frac{\nu \cdot D}{g}}\approx 0.5\mathrm{cm}$$

Of course, the condition of the negligible convection role in the heat exchange process of the microhotplate is better met, if the size of the microhotplate is(7)$$d<<4\cdot \sqrt[{3}]{\frac{\nu \cdot D}{g}}\approx 0.5\mathrm{cm}$$

This means that the majority of microhotplates fabricated using microelectronic technology satisfy this condition. In practice, the cooling of the microhotplate occurs primarily through the thermal conductivity of the surrounding air and the thermal conductivity of the supporting elements that suspend the microhotplate.

This expression defines the conditions under which only thermal conductivity and radiation should be taken into account for the estimation of heat loss of a microhotplate.

Formula (7) can be compared with the aforementioned Grashof number Gr. It is known that, for ideal gas, the values of the diffusion coefficient and kinematic viscosity coincide with each other. The analysis presented here shows that, to neglect the convective cooling of the sensor microhotplate, the value of the Grashof number of the system, including the microhotplate and surrounding gas, should be Gr << 64 or, to be more realistic, Gr << 50.

### 4.2. Convection Effects in Thermal Magnetic Oxygen Sensors

A similar analysis can be applied to the consideration of the requirements used in the design of thermal magnetic gas sensors. As previously mentioned, a very important problem is how to minimize the influence of free thermal convection in the readings of oxygen sensors.

The magnetic sensor of oxygen consists of a tube made of a diamagnetic material placed within a magnetic field of a permanent magnet. This tube is equipped with a heater and thermoanemometer for measuring gas flow through the tube. Oxygen containing gas is attracted by the magnetic field; on the other hand, magnetic susceptibility of this paramagnetic gas is inversely proportional to its temperature. Therefore, if the gas is heated within the tube, the gradient of the temperature leads to asymmetry in the system, causing a continuous flow of gas from the cooler part to the hotter part of the system.

A very important advantage of this type of gas sensor is its ability to function even in an aggressive atmosphere. However, a significant drawback is the need to maintain the orientation of the device because of the competition between the magnetic and natural convection, due to the gradient of the temperature.

To minimize this effect, it is crucial to examine the interplay between thermomagnetic and natural convection. The ideal scenario occurs when the natural convection could be entirely neglected, ensuring that the thermomagnetic convection remains independent of the orientation of the tube placed in the magnetic field. In addition, to achieve a response time sufficient for the measurement of oxygen concentration with a characteristic time of less than 10 ms, the dimension of the tube should be below a certain value defined by the diffusion processes.

Here, we will utilize the results of the previous section of this paper, which considered the role of convection in the heat exchange of the microhotplate. In that analysis, we suggested that over a microhotplate exists a virtual tube of upstreaming gas. In the case of “magnetic wind” oxygen sensors, this corresponds to a real tube with diameter d and radius r heated up to temperature T. It is evident that the most critical case, when the natural convection is most important compared to the “magnetic wind”, occurs, when the tube is oriented vertically.

Natural convection can be neglected if the time required for the gas to move through the tube due to convection (with length h) exceeds the time for the gas to diffuse back.$$\frac{h}{v}>>\frac{h^{2}}{D},\;\mathrm{or}\;v<<\frac{D}{h};$$
substituting Formula (5) for the last one, we obtain

$\frac{g\cdot r^{2}\cdot T_{av}}{8\cdot \nu }(\frac{1}{T_{r}}-\frac{1}{T})<\frac{D}{h}$, and $r^{2}h<<\frac{8D\cdot \nu }{g\cdot (\frac{T_{av}}{T_{r}}-\frac{T_{av}}{T})}$, in our case T_av_ = T, therefore(8)$$r^{2}h<<\frac{8D\cdot \nu }{g\cdot (\frac{T}{T_{r}}-1)}$$

Taking into account that both the values of the diffusion coefficient and the kinematic viscosity of air at working temperature of the magnetic oxygen sensor are equal to about 2 cm^2^/s, we can evaluate the dimension of the sensing element that would render the gas sensor insensitive to its orientation in the gravitation field and, therefore, suitable for use in portable instruments.

Let us consider the usual macroscopic gas sensor that is a glass tube equipped with a flow meter and put to a magnetic field. If the diameter of this tube is, for example, 0.1 cm, T = 2∙T_r_, the length of the tube should be by the order of magnitude h << 3 × 10^−2^/r^2^~3 cm, that is about, or less than, 0.1 cm; the total size of the sensor is, in this case, a small cylinder with a diameter and length of less than 1 mm.

It is clear that the fabrication of such a small sensor using conventional macroscopic tools is rather complicated. Therefore, a thermomagnetic oxygen sensor free of the influence of the sensor orientation should be fabricated using alternative methods, such as microfabrication, which requires extremely small and precise machinery or similar instrumentation.

## 5. Discussion

### 5.1. Microhotplates

As discussed in the Introduction, the heat exchange processes of microhotplates have been extensively described in the literature by various research groups. One of the recent examples of such a detailed consideration is presented in [20]. In this comprehensive review, the authors overviewed the application of thermoconductometric gas sensors for the analysis of gases. However, even in this article, there is no well pronounced criterion enabling the neglect of the influence of free thermal convection on the heat exchange processes in gas sensors based on thin wires or microhotplates as heating sources.

This situation is similarly observed in other publications describing the heat exchange in microhotplates used for thermocatalytic and semiconductor gas sensors. For example, in our early publications [8], we supposed that, for the rather small microhotplates with 250 × 250 μm size, the influence of convection could be disregarded. This conclusion was confirmed by the good agreement between the results obtained by the simulation of the heat exchange process and the experimental results. In both cases, we had approximately 25 mW heating power necessary to heat the sensor up to a working temperature of 450 °C used for methane detection with MOS (metal oxide semiconductor) and thermocatalytic gas sensors.

The same assumptions were applied in the design of thermocatalytic gas sensors based on microspirals and microhotplates [21]. In this study, the authors fabricated a small microhotplate based on thin anodic alumina film (thickness is of about 30 μm). The size of the hot area was of about 200 × 200 μm. Power consumption at a working temperature of 450 °C was of about 35 mW for continuous heating to this temperature.

On the other hand, today there is another tendency in gas sensor development. It consists of the analysis of multisensory responses, and one of the possible solutions is the application of multi-heater and multi-electrode sensor chips [22,23]. In cases where the sensor array is fabricated as a single microhotplate, the size of the hotplate becomes significantly larger, typically around 2 × 2 cm. In such instances, convection heat exchange processes must be taken into account.

Another scenario, in which the application of relatively big hotplates is reasonable, is the investigation of chemical, gas sensing, and other properties of sensing materials [24]. The authors of this work used a hotplate fabricated by screen printing, and the size of the hot area of this hotplate is about 6 × 3 mm (Figure 4). It consumes ~3 W at a working temperature of 350 °C.

The main reasons for using a large hotplate are to ensure uniformity of the temperature field over the hotplate and the possibility to easily deposit the sensing layer using both a screen-printing process and inkjet printing and to easily control the uniformity of the deposited sensing layer. In both last cases, the convection heat loss should be taken into account. The same situation also occurs with the application of early gas sensors manufactured by Figaro Inc., such as the TGS 812 and related models.

The application of different types of microfabricated hotplates is now widespread. In addition to the microhotplates mentioned above, it is possible to discuss commercial microhotplates fabricated by companies such as Figaro Inc. (Osaka, Japan) [25], Sensirion (Stäfa, Switzerland) [26,27], and SGX (Corcelles-Cormondreche, Switzerland) [28], and the microhotplates developed by a number of research groups. In the most part, for these microhotplates, condition (7) is met and, therefore, the free convection as a channel for heat losses can be neglected.

### 5.2. Thermomagnetic Sensors

Our colleagues [29] made an attempt to fabricate an oxygen sensor, which can be applied in portable devices (including medical instruments) using microfabrication. The sensor was fabricated as a small spiral made of 10 μm Pt glass-coated wire and the thickness of glass was of about 1 μm. The spiral diameter was about 100 μm and the length was about 150 μm. The spiral was made by winding the Pt wire around hot NiCr (Nickel-Chromium) wire followed by etching away the NiCr wire. The high temperature during the winding process facilitated the sintering of the glass insulation around the Pt wire, resulting in a structure resembling a very small glass tube containing the Pt wire spiral. The spiral was suspended on two platinum wires within a TO-18- or TO-46-type housing and placed in a magnetic field gradient. The orientation of the spiral in the magnetic field was more or less random. A schematic representation of this sensor is presented in Figure 5.

Figure 6 presents a photograph of the thermomagnetic sensor with a magnetic system consisting of two neodymium (Nd) magnets with iron magnetic conductors. The Pt wire coil is placed within the gap, and another arm with a reference coil is placed within the gap between two details made of diamagnetic Al alloy. The results of the measurement of oxygen concentration are presented in Figure 7. (This figure presents results obtained in [29]). The plot shows the potential difference between two branches of a Wheatstone bridge similar to those presented in Figure 2. The configuration of the particular microsensor is given in Figure 6. This potential difference is due to gas flow through the tubes formed by the Pt wire spiral. As previously discussed, there is a permanent gas flow of oxygen-containing gas in the gap of the permanent magnet, whereas no flow occurs in the tube positioned between the poles made of aluminum. Therefore, the spiral within the magnetic field is cooled by the gas flow, and this leads to the different resistance of the two spirals. This difference results in a potential difference between the two branches of the Wheatstone bridge, as shown in Figure 7. This potential difference is proportional to the concentration of the paramagnetic component (oxygen) in the gas-analyte.

The results presented in [29] show that the thermomagnetic sensor is capable of determining oxygen concentrations as low as approximately 0.1 vol.%. The response time of the sensor is below 0.1 s. The small size of the sensor (~10^−2^ cm, diffusion time of ~1 ms), together with the low thermal response time of the miniaturized Pt coil (~50 ms), results in a total response time of around 0.05 s. Consequently, the sensor is suitable for medical monitoring at a normal breath rate (12 min^−1^) and can even function at a high rate of up to 100 min^−1^.

Additionally, it was demonstrated that the measurement of oxygen concentration is less influenced by the orientation of the sensor compared to the macro sensor shown in Figure 1 (where the tube length is 12 mm). These findings support our conclusion regarding methods to mitigate the influence of free convection on the response of thermomagnetic oxygen sensors.

However, the sensor described in [29] requires further optimization. This concerns, first of all, the strong fixation and orientation of the microcoil with respect to the magnetic field. The suspension with two 10 μm Pt wires is insufficient for ensuring stable orientation. This issue will be addressed in future developments of the sensor, particularly in the advanced microelectronic versions of both thermomagnetic sensors and sensors based on the Senftleben effect.

## 6. Conclusions

We investigated the influence of convection on the heat exchange processes of microhotplates used in the fabrication of semiconductor and thermocatalytic (calorimetric) gas sensors, as well as the heat exchange of thermal magnetic sensors of oxygen. The analysis was based on the consideration of the competition of convection flow and back diffusion. It was shown that there is a certain critical size of the microhotplate equal to d_cr_~4·ν·Dg3, where ν is kinematic viscosity of air, D is the diffusion coefficient, g is the acceleration of free fall, and d~0.5 cm. If the size of the microhotplate d << d_cr_, the influence of convection heat exchange can be neglected and only the thermal conductivity of air and of the elements of the sensor together with the IR (infrared) radiation (if the sensor is heated up to a very high temperature) should be taken into account as channels of heat losses. This expression involves only the fundamental constants of ambient gas (air, for example). It can be compared with the Grashof number Gr = gL^3^/ν^2^, where *g* is the gravitational acceleration, *L* is the typical geometrical size of the surface, and ν is the kinematic viscosity of air, designed to form a dimensionless value, taking into account all parameters, which could control the process of convection.

Similarly, for the thermal magnetic oxygen sensor consisting of a hot dielectric tube placed within a magnetic field gradient and used to measure the concentration of paramagnetic gas oxygen, certain conditions must be met to avoid the influence of free convection on the measurement results. In this case, the convection can be neglected and, respectively, the results of the measurements of oxygen concentration should not depend on the sensor orientation, if the size of the sensor size satisfies the following condition: $r^{2}h<<\frac{8D\cdot \nu }{g\cdot (\frac{T}{T_{r}}-1)}$, where r is the radius of the dielectric tube of the sensor and h is the length of the tube.

## Figures and Tables

**Figure 1 sensors-25-02830-f001:**
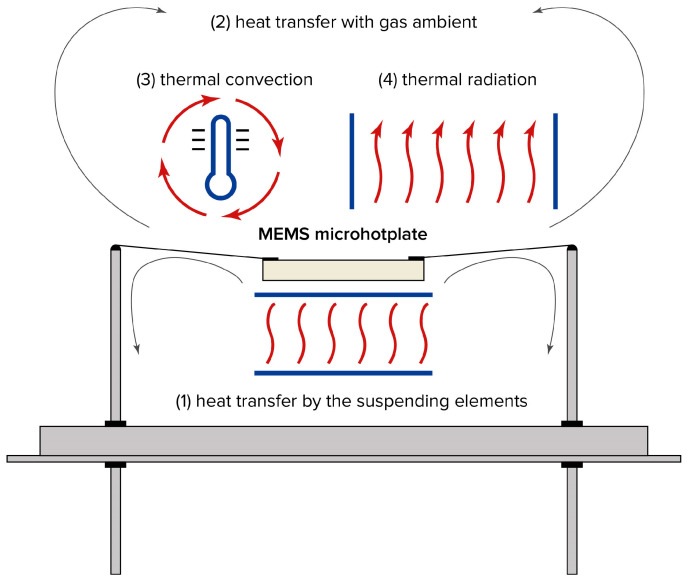
The paths of heat loss in a MEMS microhotplate, assembled in housing: (1) heat transfer by the supporting elements; (2) heat transfer due to heat conductivity of ambient gas; (3) free thermal convection; (4) thermal radiation.

**Figure 2 sensors-25-02830-f002:**
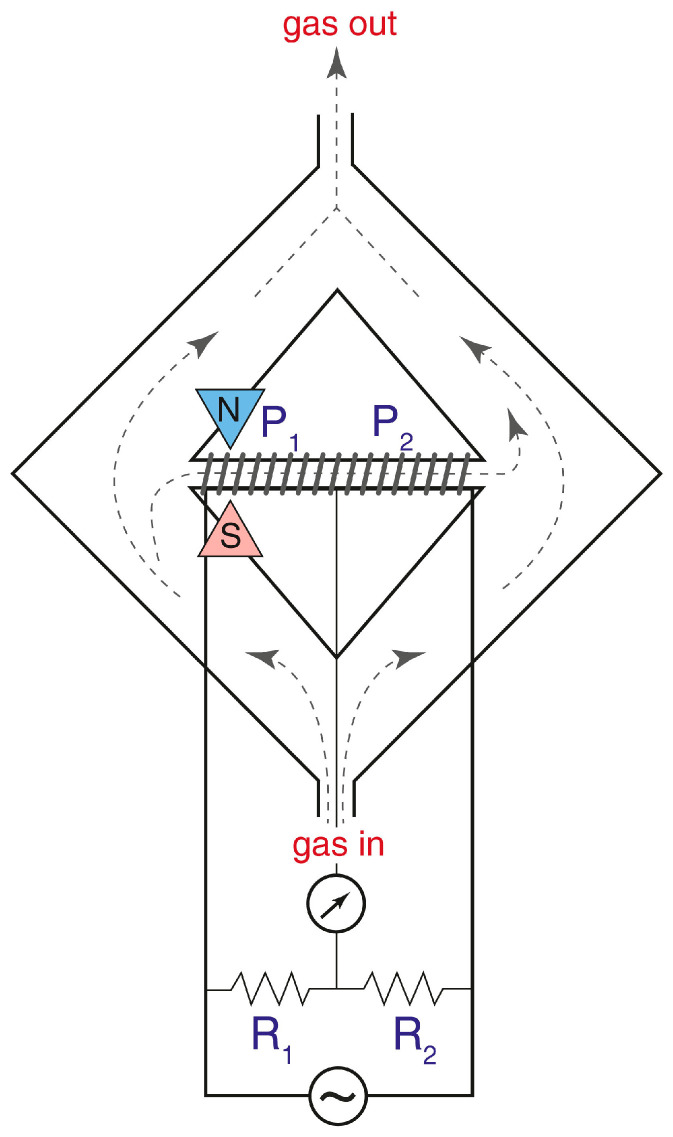
The scheme of the magnetic oxygen sensor based on the “magnetic wind” effect. The instrument consists of two branches (the left and the right ones). The gas that is to be analyzed enters from the “gas in” tube and is divided into two flows. As a result, gas pressures and velocities are the same in the left and right ends of the horizontal tube (point P_1_ and P_2_, respectively). The wire coil is made of platinum and is wound around a horizontal glass (diamagnetic) tube. This coil is used at the same time as a heater and as a thermoanemometer measuring gas velocity in the horizontal tube. This velocity is measured using a Wheatstone bridge composed of two parts of the coil and resistors R_1_ and R_2_. If the gas flows in the tube from left to right, the temperature of the left part of the coil is lower than the temperature of the right part of the coil (temperature in the circular branches is equal to room temperature). The left part of the horizontal tube is placed between the poles of the magnet, and paramagnetic oxygen is attracted by the magnetic field. Gas moving from left to right is heated gradually by the coil and loses paramagnetic susceptibility due to Curie’s law. Therefore, gas from the left side is attracted by the magnetic field more strongly than gas from the right side. This force difference assures permanent gas flow in the horizontal tube, and that the velocity of gas is proportional to the oxygen content in the gas to be analyzed.

**Figure 3 sensors-25-02830-f003:**
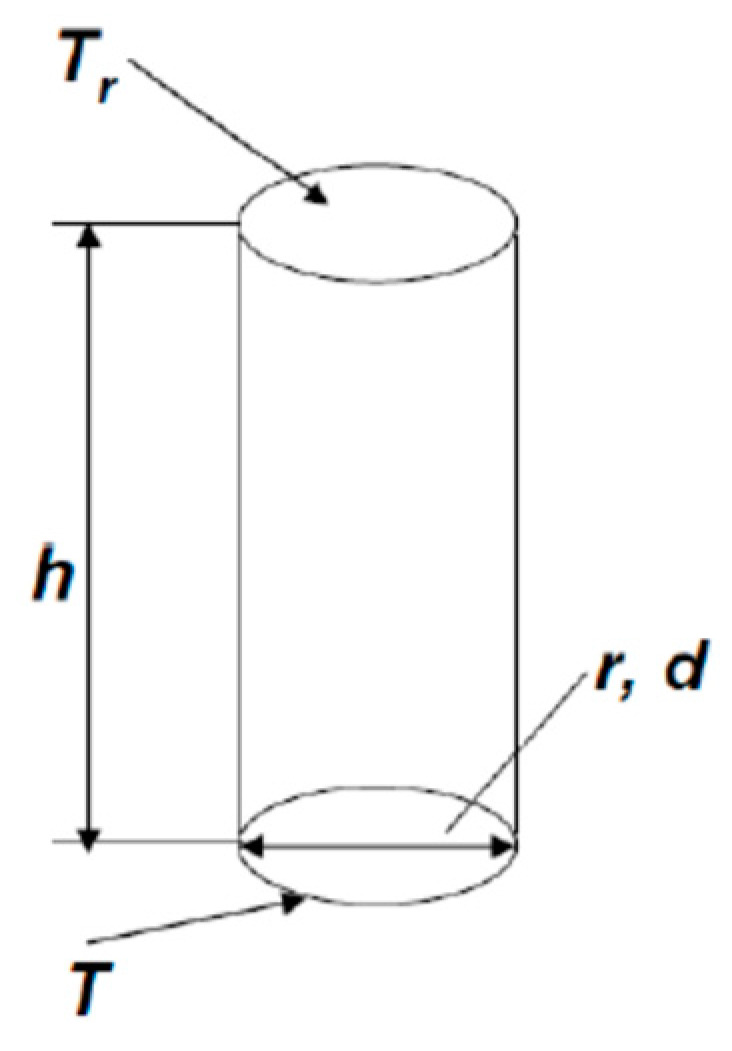
Microhotplate and convection tube over it. The microhotplateis is located in the bottom of the convection tube, the diameter and radius of the microhotplate are equal to d and r, respectively, the temperature of the microhotplate is T, and the ambient temperature is equal to T_r_. The temperature of gas surrounding the cylindrical part of convection tube is also equal to T_r_.

**Figure 4 sensors-25-02830-f004:**
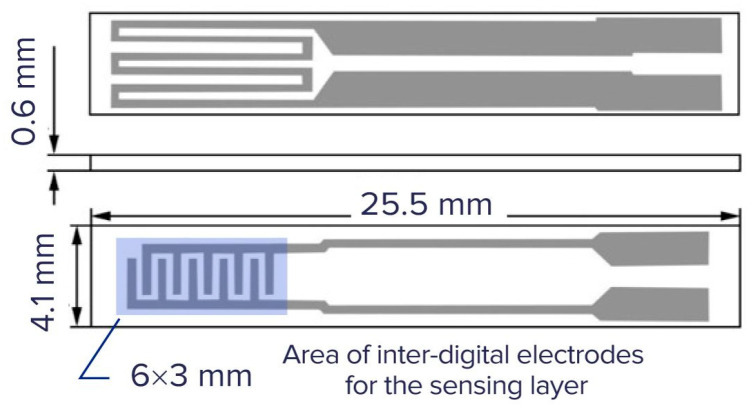
The hotplate with similar dimensions was used in [24] for the physical and chemical investigation of sensing materials.

**Figure 5 sensors-25-02830-f005:**
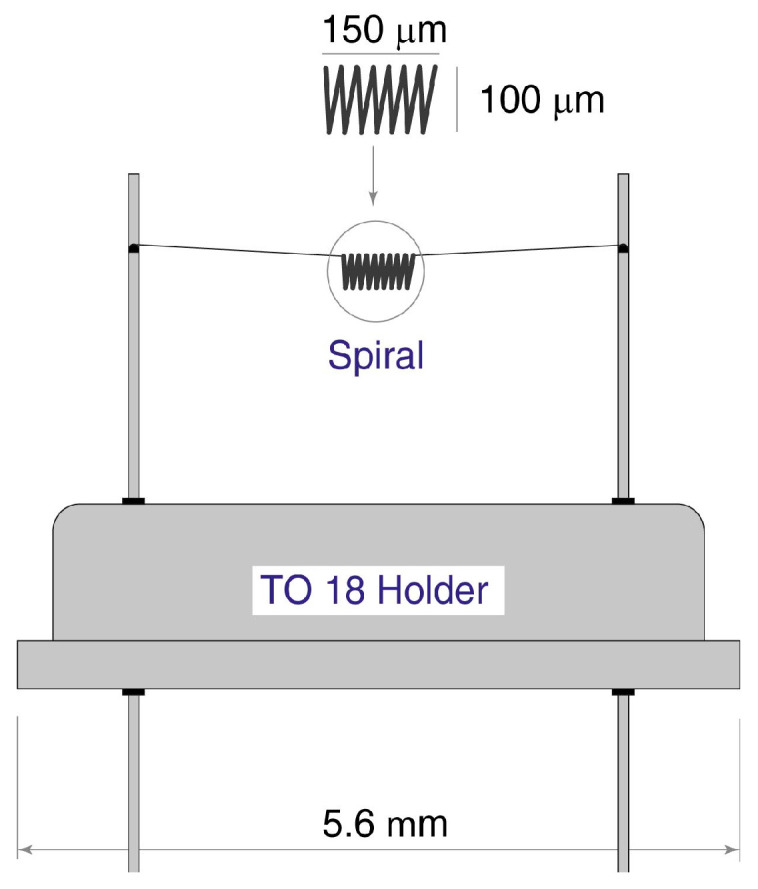
The scheme of the thermomagnetic oxygen sensor, which meets the requirements of Equation (8). The spiral is made of 10 μm platinum coated wire with a spiral size is of ~150 × 100 μm.

**Figure 6 sensors-25-02830-f006:**
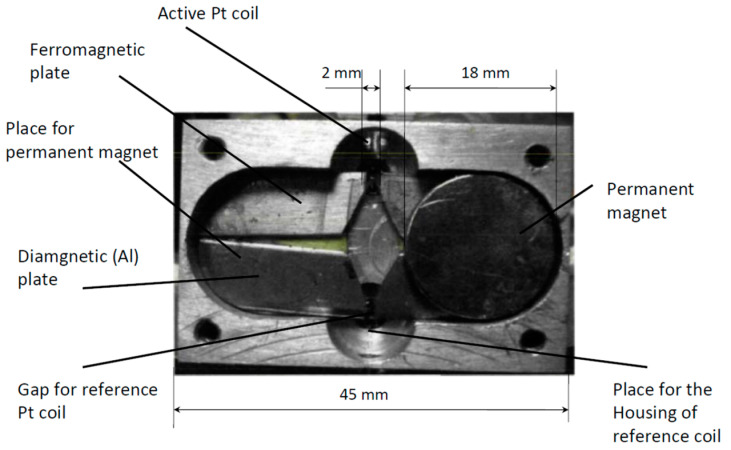
Photo of the thermomagnetic sensor [29].

**Figure 7 sensors-25-02830-f007:**
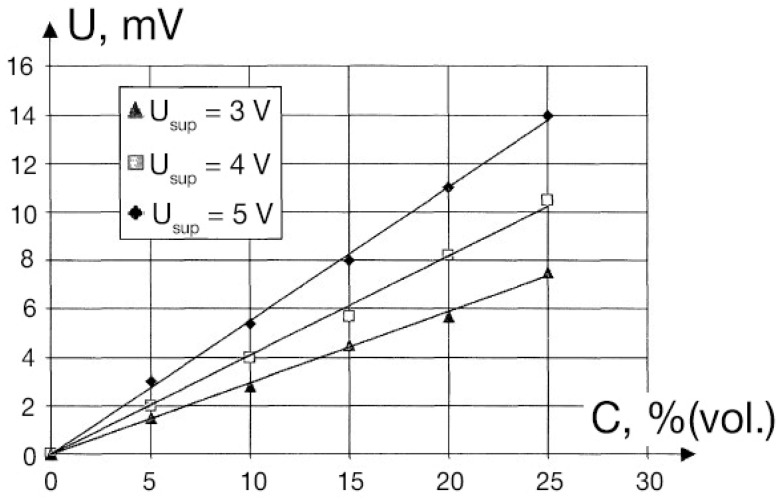
Thermomagnetic sensor response as a function of oxygen concentration in diamagnetic gas. This plot presents the potential difference between two branches of a Wheatstone bridge presented in Figure 2; the configuration of the particular sensor is given in Figure 6. The voltage indicated in the plot is a heating voltage supplied to the Wheatstone bridge heating two spirals (Figure 5) connected in series. The potential difference (Y axis) is a misbalance of two branches due to additional cooling of the spiral into the magnetic gap between the two poles of the magnet.

## Data Availability

All available data are presented in this paper.

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
