# Peer review of "The Role of Convection and Size Effects in Microhotplate Heat Exchange: Semiconductor and Thermomagnetic Gas Sensorsâ€"

_sensors, 2025, doi:10.3390/s25092830_

Round 1
Reviewer 1 Report
Comments and Suggestions for Authors
The authors analysed two different types of gas sensors namely thermocatalytic type and the oxygen sensors of thermal magnetic type. Although the theoretical aspects of the sensors are broadly studied, the manuscript requires rigorous corrections in terms of scientific presentation. There are significant areas where the manuscript is lacking with some major points, which need to be corrected and clarified.
- The introduction is too long!
- Figure 3 is missing!
- Authors need to revise the English language. The flow of the language is not rigorous in some places. For example, in section 2.1, “Semiconductor and thermocatalytic” (“Me mentioned above”), please use more passive voice to align with scientific writing conventions.
- What is “upascending”? eg, line 218.
- The authors used some figures from other publications. Those have to be mentioned as “adapted” or “modified from” with reference.
- The abstract and the first part of the conclusion sound almost the same.
- Please try to use different keywords that are not in the title (It will benefit the authors if they use the words that are not included in the title of the manuscript).
- Numerous typographical errors are scattered throughout the manuscript. Careful attention to detail is required, and English writing needs further polishing.
- References have to be homogeneous, e.g., please compare the (style) difference between references 2 and 3, 9 and 10, 23 and 24, etc. Also, there is a mistake in reference 4.
Authors need to revise the English language. The flow of the language is not rigorous in some places. Please use more passive voice to align with scientific writing conventions.
Author Response
The authors analyzed two different types of gas sensors namely thermocatalytic type and the oxygen sensors of thermal magnetic type. Although the theoretical aspects of the sensors are broadly studied, the manuscript requires rigorous corrections in terms of scientific presentation. There are significant areas where the manuscript is lacking with some major points, which need to be corrected and clarified.
The introduction is too long!
We agree with your opinion. Thank you! The Introduction part was divided into 3 separate sections.
Figure 3 is missing!
We are very sorry. There is Fig. 3 in the text of the paper. This is the scheme of the convection tube presented in p. 6 -7.
Authors need to revise the English language. The flow of the language is not rigorous in some places. For example, in section 2.1, “Semiconductor and thermocatalytic” (“Me mentioned above”), please use more passive voice to align with scientific writing conventions.
Thank you! We edited our language. “Me mentioned above” is a mistake. We wrote “It was mentioned above”.
What is “upascending”? eg, line 218.
Thank you very much! We really were not sure that this word is correct. We substituted the word “upascending air flow” by “ascending (flowing up in gravitation field) air flow”. We hope that now this is correct.
The authors used some figures from other publications. Those have to be mentioned as “adapted” or “modified from” with reference.
Thank you! We added such words.
The abstract and the first part of the conclusion sound almost the same.
We modified the abstract. Thank you!
Please try to use different keywords that are not in the title (It will benefit the authors if they use the words that are not included in the title of the manuscript).
Thank you very much! We did this.
Numerous typographical errors are scattered throughout the manuscript. Careful attention to detail is required, and English writing needs further polishing.
Thank you very much!
References have to be homogeneous, e.g., please compare the (style) difference between references 2 and 3, 9 and 10, 23 and 24, etc. Also, there is a mistake in reference 4
Thank you! We edited the style of the references. Unfortunately, we did not find any mistake in Ref. 4. Sorry!
Reviewer 2 Report
Comments and Suggestions for Authors
In the manuscript, Alexey Vasiliev el al. investigated The Role of Convection and Size Effects in Microhotplate Heat Exchange: Semiconductor and Thermomagnetic Gas Sensors. I have some minor suggestions for authors before we can proceed with a positive action.
- A well-prepared abstract enables readers to identify the basic content of a paper quickly and accurately, to determine its relevance to their interests, and thus to decide whether they need to read the paper in its entirety. Please improve.
- The results and discussion section should be discussed and compared with the previously reported results in detail.
- The clarity of Figure 7 is not sufficient
- Have the authors noticed some important works?Sensors and Actuators A: Physical 366, 2024,114954.Micro and Nanostructures 198, 2025, 208063.Journal of Alloys and Compounds 1017, 2025, 179127.
The English could be improved to more clearly express the research.
Author Response
In the manuscript, Alexey Vasiliev el al. investigated The Role of Convection and Size Effects in Microhotplate Heat Exchange: Semiconductor and Thermomagnetic Gas Sensors. I have some minor suggestions for authors before we can proceed with a positive action.
A well-prepared abstract enables readers to identify the basic content of a paper quickly and accurately, to determine its relevance to their interests, and thus to decide whether they need to read the paper in its entirety. Please improve.
Thank you! We did this.
The results and discussion section should be discussed and compared with the previously reported results in detail.
We modified the part “Results and Discussion”. Thank you!
The clarity of Figure 7 is not sufficient
We modified the text related to the Fig. 7. We hope that it became more clear.
Have the authors noticed some important works? Sensors and Actuators A: Physical 366, 2024,114954.Micro and Nanostructures 198, 2025, 208063.Journal of Alloys and Compounds 1017, 2025, 179127.
Thank you! We used of the papers mentioned by you.
Zhen Cui, Hui Wu, Kunqi Yang, Xia Wang, Yujie Lv. Adsorption of gas molecules on intrinsic and defective MoSi2N4 monolayer: Gas sensing and functionalization. Sensors & Actuators: A. Physical 366 (2024) 114954.
Reviewer 3 Report
Comments and Suggestions for Authors
Submitted manuscript is very interesting and the subject is in the hot spot of the present-day research. Work keywords are close to the MDPI Sensors keywords. However, work has severe technical deficiencies up to clear example of negligence.
All abbreviations must be defined at their first appearance (example of the problem – IR abbreviation is not defined in the abstract prior to the use).
Equations should not be used neither in the abstract, nor in conclusions part. If necessary, it should be verbal description. One of the reasons – they are not original equations of this team but referencing is absent.
Figures 2, 4 and 6-7 contain referencing to works [14, 25, 30]. However, it is not clear if formal permissions were obtained for re-use of this materials. Figure 2 must have detailed description with indication of all appearing parameters, it is useless in the present state.
Figure 5 must show the scale. It was sloppily modified and rests of previous text is clearly visible.
Figure 6 must show the scale and details of the sensor parts.
Figure 7 was prepared in foreign language different from English and it does not show either error bars or regressions for linear fit. The parameters were not defined in the text, they are not given in the figure caption. Here, one important issue appears, if all numerical results (Figure 7) are taken without permission from the work [30], what is the novelty of the present submission? Authors must provide valuable explanation.
English requires careful improvement, sometimes it is difficult to understand the exact meaning. (see example: “The results of the simulation were compared with the numerical simulation and measurement of heat losses by micromashined gas sensors. The optimal dimensions of the sensor microhotplate are given as a result of this simulation and measurements” -word simulation is badly defined here).
Comments on the Quality of English LanguageEnglish requires careful improvement, sometimes it is difficult to understand the exact meaning. (see example: “The results of the simulation were compared with the numerical simulation and measurement of heat losses by micromashined gas sensors. The optimal dimensions of the sensor microhotplate are given as a result of this simulation and measurements” -word simulation is badly defined here).
Author Response
Submitted manuscript is very interesting and the subject is in the hot spot of the present-day research. Work keywords are close to the MDPI Sensors keywords. However, work has severe technical deficiencies up to clear example of negligence.
All abbreviations must be defined at their first appearance (example of the problem – IR abbreviation is not defined in the abstract prior to the use).
Thank you! We explained the abbreviations. For example, IR means infrared.
Equations should not be used neither in the abstract, nor in conclusions part. If necessary, it should be verbal description. One of the reasons – they are not original equations of this team but referencing is absent.
We are very sorry! It is difficult to explain by words even this simple equation. In addition, this is an original equation of the team. In reality, this is one of the main results of the work.
Figures 2, 4 and 6-7 contain referencing to works [14, 25, 30]. However, it is not clear if formal permissions were obtained for re-use of this materials. Figure 2 must have detailed description with indication of all appearing parameters, it is useless in the present state.
Thank you very much! Usually, the formal permissions for the re-use of pictures are requested from the Editor after the acceptance of the paper for publication. As we know, this is a rule of MDPI. Please let us know if this is not correct. Reference [30] in old version of the paper (reference [31] in actual one) is an abstract of PhD thesis. There is no formal editor of this document; it was published as a manuscript. This status of the document is indicated in this PhD abstract [31]. We hope that this picture can be re-used with appropriate reference.
We added detailed description of the thermomagnetic sensor operation principle to the caption of the Fig. 2.
Figure 5 must show the scale. It was sloppily modified and rests of previous text is clearly visible.
Your are right. This was scan from of picture taken from Ref 31, and the text from the next page is visible through paper. Sorry! We substituted the picture. As well, we showed the scale of the sensor element.
Figure 6 must show the scale and details of the sensor parts.
We modified Fig. 6.
Figure 7 was prepared in foreign language different from English and it does not show either error bars or regressions for linear fit. The parameters were not defined in the text, they are not given in the figure caption. Here, one important issue appears, if all numerical results (Figure 7) are taken without permission from the work [30], what is the novelty of the present submission? Authors must provide valuable explanation.
Thank you very much for valuable remark! We edited the text in Russian, which appeared by our mistake in the plot. Sorry!
The results of the paper are summarized in the conclusion. The main result is the clear criteria of the significance (or negligibility) of the convection effects in the heat exchange of microhotplates of thermocatalytic and semiconductor gas sensors and of thermomagnetic oxygen sensors.
Therefore, the results presented in Fig. 7 are used only for the illustration of the results of our research. This is not a result of our investigation.
Comments on the Quality of English Language
English requires careful improvement, sometimes it is difficult to understand the exact meaning. (see example: “The results of the simulation were compared with the numerical simulation and measurement of heat losses by micromashined gas sensors. The optimal dimensions of the sensor microhotplate are given as a result of this simulation and measurements” -word simulation is badly defined here).
Thank you very much! We corrected the text as much as we can. The paragraph mentioned by you is really bad. Sorry!
Round 2
Reviewer 1 Report
Comments and Suggestions for Authors
The authors have appropriately addressed all comments and questions. Regarding Ref. 4, removing the extraneous number (“6”) has corrected the issue, and it now appears appropriate in the revised manuscript.
However, in response to the reviewer’s comment regarding the absence of Figure 3, the authors stated: “There is Fig. 3 in the text of the paper. This is the scheme of the convection tube presented in p. 6 -7. Despite this clarification, I am still unable to locate Figure 3 in the version of the manuscript currently available to me. While there are captions for Figure 3 at line 283 in the revised version and at line 250 in the earlier version, the figure itself appears to be missing in both.
If the authors intend to refer to a modified version of a previous figure as Figure 3 without including the actual figure, it may lead to confusion and hinder the readers' understanding.
Other than this, in my view, the revised manuscript now satisfies the standards required for publication in Sensors.
Author Response
We would like to thank you for the careful analysis of our paper. We met a very strange effect, Fig. 3, which existed in our manuscript, disappeared in the Word file after uploading to the website of the MDPI Sensors. We cannot explain what the reason of this effect is. We changed the format of the picture, and hope that this time everything will be OK.
We hope that this time, in a new version of the paper Fig. 3 will not disappear again. We will check this issue.
We attach pdf version of the paper.
Thank you once more!

Reviewer 2 Report
Comments and Suggestions for Authors
accept
Author Response
Dear Sir/Madam
We would like to thank you for the assistance!
Reviewer 3 Report
Comments and Suggestions for Authors
Submitted manuscript is very interesting and the subject is in the hot spot of the present-day research. Work keywords are close to the MDPI Sensors keywords. However, work has technical deficiencies.
Equations should not be used neither in the abstract, nor in conclusions part. If necessary, it should be verbal description. One of the reasons – they are not original equations of this team but referencing is absent.
If referencing for figure is used, permission should be given. If the work is published in non-indexed source the referencing is not necessary. I believe that Editor will make a decision with this respect.
English requires careful improvement and I propose to address this issue to the MDPI services.
Comments on the Quality of English Language
Submitted manuscript is very interesting and the subject is in the hot spot of the present-day research. Work keywords are close to the MDPI Sensors keywords. However, work has technical deficiencies.
Equations should not be used neither in the abstract, nor in conclusions part. If necessary, it should be verbal description. One of the reasons – they are not original equations of this team but referencing is absent.
If referencing for figure is used, permission should be given. If the work is published in non-indexed source the referencing is not necessary. I believe that Editor will make a decision with this respect.
English requires careful improvement and I propose to address this issue to the MDPI services.
Author Response
Thank you for this important comment. To clarify this issue, we asked Ms. Yuliya Yu, Assistant Editor of the Special Issue, about the opinion of the MDPI Sensors. Here is her answer, received April, 17, 2025:
“Dear Professor Vasiliev,
Thank you for your email.
We would like to kindly explain that for the manuscript. Sensors format
permits the use of short formula in the abstract and conclusion
sections. You could mention this in your response.
If you have any concerns, please do not hesitate to contact us.
Best Regards,
Ms. Yuliya Yu
Assistant Editor
Sensors (https://www.mdpi.com/journal/Sensors/)”
Therefore, we are very sorry, but we will keep our short formula in the Abstract and in the Conclusion sections according to the recommendations of the MDPI Sensors.
If referencing for figure is used, permission should be given. If the work is published in non-indexed source the referencing is not necessary. I believe that Editor will make a decision with this respect.
Thank you! We also believe that Editor will make a decision with this respect.
English requires careful improvement and I propose to address this issue to the MDPI services.
We improved the language as much as it was possible. Thank you!